# Integrated Clinical, Molecular and Immunological Characterization of Pulmonary Sarcomatoid Carcinomas Reveals an Immune Escape Mechanism That May Influence Therapeutic Strategies

**DOI:** 10.3390/ijms241310558

**Published:** 2023-06-23

**Authors:** Susann Stephan-Falkenau, Anna Streubel, Thomas Mairinger, Torsten-Gerriet Blum, Jens Kollmeier, Fabian D. Mairinger, Torsten Bauer, Joachim Pfannschmidt, Manuel Hollmann, Michael Wessolly

**Affiliations:** 1Institute for Tissue Diagnostics, MVZ at Helios Klinikum Emil von Behring, 14165 Berlin, Germany; anna.streubel@helios-gesundheit.de (A.S.); thomas.mairinger@helios-gesundheit.de (T.M.); manuel.hollmann@helios-gesundheit.de (M.H.); 2Department of Pneumology, Heckeshorn Lung Clinic, Helios Klinikum Emil von Behring, 14165 Berlin, Germany; torsten-gerriet.blum@helios-gesundheit.de (T.-G.B.); jens.kollmeier@helios-gesundheit.de (J.K.); torsten.bauer@helios-gesundheit.de (T.B.); 3Institute of Pathology, University Hospital Essen, University of Duisburg-Essen, 45147 Essen, Germany; fabian.mairinger@uk-essen.de (F.D.M.); michael.wessolly@uk-essen.de (M.W.); 4German Cancer Consortium (DKTK), Partner Site University Hospital Essen, 45147 Essen, Germany; 5Department of Thoracic Surgery, Heckeshorn Lung Clinic, Helios Klinikum Emil von Behring, 14165 Berlin, Germany; joachim.pfannschmidt@helios-gesundheit.de

**Keywords:** pulmonary sarcomatoid carcinoma, PSC, NSCLC, precision oncology, targeted therapy, immunotherapy, MET Exon 14 skipping mutations, PD-L1 expression, processing escape

## Abstract

Pulmonary sarcomatoid carcinoma (PSC) has highly aggressive biological behaviour and poor clinical outcomes, raising expectations for new therapeutic strategies. We characterized 179 PSC by immunohistochemistry, next-generation sequencing and in silico analysis using a deep learning algorithm with respect to clinical, immunological and molecular features. PSC was more common in men, older ages and smokers. Surgery was an independent factor (*p* < 0.01) of overall survival (OS). PD-L1 expression was detected in 82.1% of all patients. PSC patients displaying altered epitopes due to processing mutations showed another PD-L1-independent immune escape mechanism, which also significantly influenced OS (*p* < 0.02). The effect was also maintained when only advanced tumour stages were considered (*p* < 0.01). These patients also showed improved survival with a significant correlation for immunotherapy (*p* < 0.05) when few or no processing mutations were detected, although this should be interpreted with caution due to the small number of patients studied. Genomic alterations for which there are already approved drugs were present in 35.4% of patients. Met exon 14 skipping was found more frequently (13.7%) and EGFR mutations less frequently (1.7%) than in other NSCLC. In summary, in addition to the divergent genomic landscape of PSC, the specific immunological features of this prognostically poor subtype should be considered in therapy stratification.

## 1. Introduction

Immunotherapy and targeted therapy have brought treatment options for non-small cell lung cancer into a new era with remarkable success. However, despite new treatment strategies, lung cancer leads the statistics of cancer-related deaths in both men and women, accounting for 20.4% of all cancer deaths [1]. Within non-small cell lung cancer, the subgroup of pulmonary sarcomatoid carcinoma (PSC) has an even more aggressive clinical course, associated with a poorer prognosis and shorter survival compared to other NSCLC subtypes [2,3,4]. PSCs are a heterogeneous group of tumours that are predominantly poorly differentiated [3]. According to the current WHO classification [5], PSCs are divided into five histological subtypes: (1) pleomorphic carcinoma, (2) spindle cell carcinoma, (3) giant cell carcinoma, (4) carcinosarcoma and (5) pulmoblastoma. Among these, pleomorphic carcinoma is the most common and also presents with the worst prognosis [6]. Pulmonary sarcomatoid carcinomas are rare and are reported to have an incidence of approximately 0.4–0.5% within NSCLC [3,4]. However, diagnosis is particularly difficult with small biopsies [7], as sarcomatoid differentiation according to the WHO classification can only be suspected on biopsy-derived tumour tissue, but not definitively established. It is therefore to be assumed that the prevalence is higher than documented, especially in advanced tumour stages where the tumour entity cannot be validated on the surgical specimen.

Surgical therapy is the treatment of choice for patients in early tumour stages I–IIIA [8,9] and has been shown to result in better outcomes [10,11,12], e.g., longer overall survival (HR 0.40, *p* < 0.001) compared to patients who did not undergo surgery [4]. For patients in the advanced stages IIIB–IV, surgery is no longer suitable. These patients are treated according to the guidelines that apply to NSCLC in general [8,9]. However, in contrast to other NSCLC subtypes, PSCs are considered chemo- and radiation-refractory [13,14,15], with randomized controlled trials exploring systemic therapies for advanced PSC being scarce [16]. Furthermore, the more aggressive biological behaviour and poorer prognosis of PSCs seems to also be driven by specific tumour characteristics in the cellular immune defence and at the molecular level that need to be taken into account.

Thus, programmed cell death ligand-1 (PD-L1) expression appears to be high in PSC [17,18,19,20,21,22] and it has been demonstrated that PD-L1 expression in PSC patients is related to aggressive pathological features such as ipsilateral mediastinal and/or subcarinal lymph node involvement (pN2) and metastasis, and is also slightly correlated with a lower probability of overall survival and disease-free survival (*p* = 0.069 and *p* = 0.015, respectively) [22]. As PSC is generally rare, it should be noted that these studies only evaluated PD-L1 status in a relatively small number of patients (No. of patients 13 to 75).

A potential biomarker for immunotherapy also appears to be the tumour mutational burden (TMB). It is assumed that a high TMB leads to an increased presentation of neo-antigens on the surface of the tumour cells. These are recognized by the immune system and the tumour cells are consecutively eliminated by the cellular immune defence [23,24]. Zhou et al. reported a high TMB in 58 patients with pure PSC showing a median TMB of 8.6 mutations/Mb whereby 29/58 (50%) of cases had a TMB > 20 mutations/Mb [17]. Both PD-L1 and TMB are considered independent predictive of immunotherapy with immune checkpoint inhibitors (ICIs) such as CTLA-4 or PD-1/PDL-1 antibodies. However, there are also PSCs with high PD-L1 expression or high mutational load that do not respond to therapy with ICIs [25].

Recent approaches to explain other immune escape mechanisms explore the concept of altered processing of small peptide fragments that are presented on the cell surface (epitopes), which are the key targets for cellular defence by CD8-positive cytotoxic T-lymphocytes (CTLs). The deposition of epitopes on the cell surface requires a complex intracellular process involving proteasome degradation, and TAP- as well as HLA-binding [26,27,28,29,30]. The mechanisms of immune escapes caused by a deficient antigen presentation have been well described in various tumours, including an impaired function of proteasomal degradation [31,32] and the TAP-transporter [31] or the deficiency of HLA/MHC class I molecules due to point mutations or large deletions [30,33,34]. However, the majority of tumours may not even harbour disruptive HLA alterations. HLA/MHC presentation is negatively impacted, regardless [35,36]. For example, this can be achieved via epigenetic silencing, mediated by NLRC5 [37] or EZH2 [38]. Furthermore, miRNAs may also mediate mRNA degradation of MHC class I or associated factors post transcription [39,40]. Additionally, transcription factors like E2F1 repress the expression of MHC/HLA class I associated components [41]. Lastly, translational regulators, e.g., MEX3B are implicated in downregulation of HLA class I and mediation of therapy resistance [42].

Regarding impaired proteasomal degradation, it is not the epitope itself but mainly the flanking regions that are held responsible for the peptide not reaching the optimal size for sufficient presentation, resulting in a less effective T cell response [43]. Thus, insufficient or length-altered epitope representation due to poor processing could explain why some tumours do not respond to immunotherapy. This assumption is supported by studies of viral infections with the human immunodeficiency virus 1 (HIV-1) and the hepatitis C virus (HCV), in which a subset of specific mutations altered the proteasomal processing of the viral proteins. These mutations led to altered epitopes with different lengths and to a reduced activation of cytotoxic T cells. In the present study, these mutations are referred to as processing mutations [44,45]. Using deep learning algorithms, we have previously been able to demonstrate that NSCLCs exhibit a subset of non-synonymous mutations (processing mutations). Most processing mutations are derived from passenger alterations, rather than driver alterations [46]. These mutations lead to altered epitope processing by the proteasome. Epitopes are processed differently due to the accumulation of mutations, which change cleavage patterns of proteasomal β-subunits that prefer to cleave after specific amino acid residues [47,48]. Altered epitopes were less efficient than their wildtype state in triggering an immune response, thereby leading to an immune escape under therapeutic pressure with ICIs, which in turn is associated with a shortened overall survival time [49,50].

In addition, PSCs display differences in the frequency and distribution of oncogenic driver mutations compared to other NSCLCs. For example, an increased occurrence of MET exon 14 skipping mutations has been found in larger Caucasian and Chinese cohorts [51,52], which lead to a splice variant of the MET gene with increased activation of MET kinase and increased tumour growth. Tong et al. showed that MET exon 14 skipping is an independent prognostic factor associated with poorer survival in multivariate analysis [52]. In contrast, other targetable mutations, e.g., in the in EGF-receptor gene or ALK or ROS1 translocations, may occur less frequently.

Although there are several studies on clinicopathological and molecular features as well as prognosis of PSC [6,11,13,17,51,53] the study populations are mostly small or related to different ethnicities [12]. PSCs are still poorly understood to date. Characterizing the complex interlacement of their particular immune-escape mechanisms and molecular profile is therefore urgently needed to increase our understanding of the pathogenesis of these tumours and pave the way for the development of new biomarkers. The aim of the study was to investigate the immunological and molecular features of PSC in what is, to our knowledge, the largest cohort of 179 patients of Caucasian descent. In particular, we wanted to explore the different immune escape mechanisms like PD-L1 expression and altered epitope processing. Do processing mutations also occur in PSC and do they have an impact on overall survival? We were also interested in whether the type of therapy is related to the mechanisms of altered processing and how oncogenic driver mutations occurring in PSCs should be evaluated in this context. Finally, we aimed to describe the genomic landscape of pulmonary sarcomatoid carcinomas with regard to their potential for targeted therapy.

## 2. Results

### 2.1. Patient Characteristics

Between January 2006 and September 2022, 179 patients were diagnosed with pulmonary sarcomatoid carcinoma at the Department of Pathology, Lung Cancer Center, Helios Klinikum Emil von Behring (HKEvB), Berlin, Germany. All patients were of Caucasian descent. Overall, 125/179 (69.8%) were older than 60 years. The majority were men 105/179 (58.7%) compared to women 74/179 (41.3%). The most common subtype was pleomorphic carcinoma, 116/179 (64.8%), followed by spindle cell carcinoma, 42/179 (23.5%) and giant cell carcinoma, 16/179 (8.9%). The carcinosarcoma was the rarest tumour with 5/179 (2.8%), and a pulmonary blastoma was not diagnosed. The vast majority of tumours 178/179 (99.4%) were poorly differentiated (G3) or undifferentiated (G4). A complete TNM-stage was determined in 177/179 (98.9%) patients. At initial diagnosis, 74/177 (41.2%) of all patients already had distant metastases. Slightly more than a half of all patients 90/179 (50.8%) were diagnosed to be in advanced stages IIIB/IV, all shown in Table 1.

Of 179 patients, 135 (75.4%) had a documented history of smoking. Of these, 115/135 (85.2%) were former or active smokers at the time of diagnosis.

### 2.2. Programmed Death Cell-Ligand 1 (PD-L1) Expression in Pulmonary Sarcomatoid Cancer

Of all sarcomatoid NSCLCs, 147/179 (82.1%) presented a positive expression of PD-L1 on the tumour cell surface (TPS ≥ 1). The majority of patients, 106/179 (59.2%), showed high expression with a TPS score of ≥50%. Only in 32/179 (17.9%) cases, the tumour cells showed no or less than 1% PD-L1 expression. (Figure 1).

Interestingly, sarcomatoid carcinomas in early tumour stages I–IIIA tend to show more often no or less than 1% PD-L1 tumour cell expression (20.7%; 18/87), more tumours (26.4%; 23/87) with lower expression (>1–49%) and slightly fewer tumours (55%; 48/87) with high PD-L1 expression (≥50%) than in advanced stages IIIB–IV.

### 2.3. Distribution of Genomic Alterations in Sarcomatoid Lung Cancer

Of all patients eligible for molecular testing, 175/179 (97.8%), the vast majority, 155/175 (88.6%), showed one or more genomic alteration (Figure 2). Only 20/175 (11.4%) of patients had no detectable mutations or fusions in the analysed genes. By far the most frequent mutations were detected in TP53 (111/175, 63.4%), of which 34/175 (19.4%) were sole mutations, the remaining being found as co-occurring mutations with other drivers.

### 2.4. Genomic Alterations in PSC with Currently Approved Therapy Options

Oncogenic driver alterations with currently approved treatment options (EGFR, KRAS G12C, BRAF V600E, ALK, RET, METEx14skip) were found in 62/175 (35.4%) of patients, as shown in Figure 3. Interestingly, the frequency of activating EGFR mutations, detected in 3/175 (1.7%) patients is much lower than in other subtypes of NSCLC, e.g., adenocarcinoma of the lung (ACA) with an incidence of 12.7% reported in our previous work [54]. Mutations in ERBB2, frequently found in adenocarcinomas were not detected in PSC, whereas mutations in KRAS 59/175 (33.7%) and BRAF 7/175 (4.0%) are comparable to those in ACA [54]. Most interesting, however, is the high rate of PSCs harbouring MET mutations in 24/175 (13.7%) with consecutive MET exon 14 skipping events. In our previous work mentioned above, the detected frequency for MET Exon 14 skipping mutations in adenocarcinomas was significantly lower at 2.29%.

### 2.5. Mutations and Gene Rearrangements in PSC with high PD-L1 Expression

Of particular interest is that 93/102 (91.2%) of PSC with high PD-L1 expression (Tumour Proportion Score, TPS > 50%) also revealed genomic alterations. Most common were variants in TP53, which occurred either singly or, in most cases, as co-mutations. However, 34/102 (33.3%) of patients were also found to have mutations and fusions with already approved treatment options as shown in Table 2. For example, 19/44 (43.2%) of all KRAS mutations detected in patients with high PD-L1 expression have been the actionable mutation p.G12C. Interestingly, 12/102 (11.8%) of all PSC with high PD-L1 expression are associated with a MET exon 14 skipping event. In addition, we found one translocation resp. rearrangement in ALK (EML4-ALK; E13A20) and RET (CCDC6-RET).

Looking at the patients with treatable oncogenic alterations, 19/25 (76%) of patients with KRAS p.G12C and 12/24 (50%) of patients with MET Exon 14 skipping events have high PD-L1 expression (≥50%), as do the individual patients with BRAF p.V600E mutation and ALK, respectively, RET rearrangements mentioned above. The detailed data for all TPS scores associated with targetable oncogenic driver mutations are presented in Appendix A.

### 2.6. Processing Escapes

#### 2.6.1. Processing Escapes Are Common in PSC

In total, 302 non-synonymous mutations (Mutational Load) were identified by targeted-panel sequencing. It should be noted that the Mutational Load does not reflect non-synonymous mutations in general, but a small subset of them (missense mutations). Based on NetChop and NetMHC analysis, 146/302 (48.3%) of these mutations were revealed to influence epitope processing (Figure 4).

Looking at the distribution of non-synonymous mutations across genes covered by the used gene panel, most of them were found in TP53, KRAS, MET, SMAD4 and DDR2 (Figure 5). Processing mutations were mostly retained in TP53 and SMAD4. Strikingly, processing mutations were largely absent from other drivers like KRAS, STK11 or MET, though these genes displayed a moderate amount of non-synonymous mutations (<10 mutations).

A total of 146 processing mutations resulted in 3623 altered epitopes, including transcript variants. Overall, 2610/3623 (72%) epitopes were no longer bound by known MHC class I supertypes (4.5.3.3.). While 1013/3623 (28%) altered epitopes were still bound by MHC class I, they were unable to initiate an immune response.

#### 2.6.2. Processing Mutations Correlated with OS in Advanced-Stage PSC

Multiple covariates were correlated with overall survival (OS): Processing mutations (two variants), PD-L1 expression, Mutational Load as well as applied therapy regimens (chemotherapy and immunotherapy). Patients could display either a high or low mutation count. Patients with more than one mutation were grouped as “high”, while patients with exactly one mutation or no mutation were grouped as “low” or “none”, respectively. Of all covariates, the application of chemotherapy, receiving surgery as well as a low Mutational Load was linked to significantly reduced risk of death according to the hazard rations displayed (Figure 6a, *p* < 0.05, score-logrank test). Processing mutations also displayed a significant influence on OS (*p* = 0.0137, score-logrank test). It should be noted that patients without processing mutations were included into the patient group with a low mutation count for ease of observation in Figure 6. Processing mutations were the only significant variable that retained their influence on OS in a subgroup analysis, only concerning advanced stage patients (TNM stage IIIB–IV).

Concerning this subgroup, the influence on OS by processing mutations was independent of Mutational Load, any form of therapy or PD-L1 expression (*p* > 0.05, likelihood-ratio test, score-logrank test and Wald test), indicating processing mutations as an influential factor regarding OS estimations (Figure 6b). PD-L1 expression itself displayed no significant influence on OS in univariate or multivariate analyses. The amount of processing mutations had a definite influence on OS (*p* < 0.05, likelihood-ratio test, score-logrank test and Wald test), with patients harbouring a high mutation count (>1) also being linked to a worse prognosis (Appendix A). This influence is retained even when including patients not displaying any processing mutations.

#### 2.6.3. Processing Mutations Are Stronger Represented within Pulmonary Sarcomatoid Carcinomas Compared to Other NSCLC Subtypes

Based on previous works, we intended to compare the abundance of processing mutations between different NSCLC subsets. For this purpose, we utilized a previously established cohort of 48 NSCLC patients (ICB cohort, Section 4.5.1). In contrast, those patients were diagnosed with lung adenocarcinomas and lung squamous-cell carcinomas without sarcomatoid differentiation. They also received a wide array of different therapeutic regimens including chemo-, radiation- and immunotherapy, making it a viable control group. However, the variance in the sample size should also be noted (48 vs. 145).

First, we compared the density distribution of samples around specific mutation counts (Figure 7). In the ICB cohort, most patients peaked around zero mutations, with a lower maximum around one mutation (median amount of processing mutations). In contrast, the study cohort displayed three significant maxima at 0, 1 and 2. This may hint that processing mutations are somewhat more represented in the sarcomatoid (SARC) NSCLC cohort.

Furthermore, we employed the usage of a double-dichotomous contingency table and the Fisher’s exact test for a statistical comparison. Resulting from this correlation analysis, the chance to encounter a patient with processing mutations was increased by a factor of 2.85 (Odds ratio, OR) in the SARC NSCLC cohort (*p* < 0.05, Fisher’s exact test). In order to avoid a bias due to the lower sample size (145 vs. 48), we also compared the SARC cohort with another regular NSCLC cohort (TCGA cohort, Section 4.5.1), also mentioned in previous works [49,50]. However, this cohort was provided by The Cancer Genome Atlas (TCGA) website and dated back to 2012/2013, before immunotherapy was established in NSCLC. Though, the sample size is much higher (408 patients). Comparing SARC NSCLCs to the TCGA-NSCLC cohort, processing mutations were again more likely encountered in the SARC NSCLC cohort (OR 3.05, *p* < 0.05, Fisher’s exact test).

#### 2.6.4. Patients with Processing Mutations and Advanced-Stage Pulmonary Sarcomatoid Carcinoma Display a Worse Survival Prognosis

Data from the SARC NSCLC cohort (advanced stages IIIB–IV only) and ICB cohort (Figure 7) were pooled to detect survival differences based on processing mutations and tumour type (other subtypes of NSCLC vs. SARC NSCLC). In this analysis, both the sarcomatoid subtype and the presence of processing mutations were linked to impaired OS (*p* < 0.05, likelihood-ratio test, score-logrank test, Wald test). Moreso, the course of survival was significantly worse in SARC NSCLCs with processing mutations, compared to regular NSCLCs and processing mutations (Figure 8, *p* < 0.05, score-logrank test). In multivariate analysis, OS was seemingly more strongly influenced by the more aggressive sarcomatoid subtype (*p* < 0.05, likelihood-ratio test, score-logrank test, Wald test).

#### 2.6.5. Therapy in Patients with Few or None Processing Mutations Is Associated with Improved Survival Outcomes

As previously shown, the application of chemotherapy significantly correlates with a reduced risk of death (Figure 6). However, the effect of chemotherapy on OS is not retained when considering the subgroup of advanced tumour stages separately (*p* = 0.12, score-logrank test). In contrast, immunotherapy does not display a significant influence (*p* > 0.05, score-logrank test). However, the results from this analysis may be volatile since only 10 of 145 patients received immunotherapy (mostly pembrolizumab). Only four patients received mono-immunotherapy, without the addition of chemotherapy. Interestingly, in multivariate analysis, immunotherapy seems to display survival benefits in patients with low processing mutations (*p* < 0.05, likelihood-ratio test, score-logrank test, Wald test) (Appendix A).

## 3. Discussion

Pulmonary sarcomatoid carcinomas are characterised by a more aggressive biological behaviour and a poorer prognosis with shorter overall survival compared to other non-small cell lung carcinomas. They differ from other NSCLCs in their uncertain response to chemotherapy and radiotherapy, but also in particular immunological and molecular characteristics that may offer approaches to new therapeutic options. In this retrospective single centre observational study, we describe the largest collective of Caucasian patients to our knowledge from 179 primarily diagnosed pulmonary sarcomatoid carcinomas with regard to their complex immunological and molecular characteristics.

In summary, we find a subgroup of non-small cell lung carcinomas occurring particularly in men and at older ages, as well as in smokers. Pleomorphic carcinoma was the most common subtype. Almost all PSC examined displayed a poor histological grading (G3/G4). Slightly more than half of the patients (90/179; 50.8%) are already in an advanced tumour stage at the time of diagnosis, necessitating a palliative systemic therapy approach. In the multivariate analysis of all cases studied, surgery (*p* < 0.05, score-logrank test), chemotherapy (*p* < 0.05, score-logrank test) and Mutational Load (*p* < 0.05, score-logrank test) are shown to be independent factors of overall survival, associated with a significant reduction in risk of death. Nevertheless, these effects are not maintained when only patients in advanced tumour stages (IIIB–IV) are considered. In 147/179 (82.1%) of patients with PSC, positive PD-L1 expression can be detected on the tumour cells. Approximately three times more frequently (OR 2.85–3.05, *p* < 0.05, Fisher’s exact test) than in other NSCLC subtypes, patients with PSC show an additional, PD-L1-independent immune escape mechanism. In these tumours, processing mutations occur that lead to altered processing of epitopes, which are either no longer presented by HLA/MHC class I or are still presented but no longer immunogenic, resulting in impaired T-cell-mediated autoimmune defence. Patients with only a few or no processing mutations display a significantly lower risk of death (*p* = 0.0137). The effect is also maintained when only PSC patients in advanced tumour stages were considered (Figure 6B). Advanced-stage patients with few or no processing mutations also show improved survival depending on the therapy used, with a significant correlation for immunotherapy. Furthermore, we find an accumulation and distribution of genomic alterations that deviates from other subgroups of NSCLC, such as an increased incidence of MET exon 14 skipping events and much less activating mutations in EGFR. In 61/175 (34.9%) of patients with sarcomatoid lung carcinomas, genomic alterations are found for which approved drugs are available. A further 20% of patients with PSC have mutations for which there are no options for targeted therapy as of yet, e.g., KRAS non-G12C mutations. The majority of all cases with driver mutations also express PD-L1. Of greatest interest is also that tumours with oncogenic driver mutations are largely free of processing mutations, which might be helpful to aid therapy decisions in favour of immunotherapy, regardless of PD-L1 expression, as long as no targeted therapy is indicated.

Pulmonary sarcomatoid carcinomas are rare non-small cell lung carcinomas with an incidence of approximately 0.4–0.5% [3,4]. Diagnosis can be particularly difficult on small biopsy samples, as the current WHO classification requires at least 10% of spindle cells or pleomorphic differentiated tumour cells in the total tumour volume [5]. However, slightly more than half of the patients (90/179; 50.8%) are already in the advanced tumour stages IIIB–IV at initial diagnosis, which are no longer amenable to surgical therapy and thus to an assessment of total tumour volume on the surgical specimen. There is therefore a risk of underestimating the incidence of sarcomatoid lung carcinomas [13,55]. In our own retrospective study of 2066 patients with non-small cell lung cancer (NSCLC), the incidence of PSC was 2.6%, including biopsy specimens that showed various aspects of sarcomatoid differentiation (spindle cell or pleomorphic morphology, co-expression of the mesenchymal marker vimentin with high proliferative activity) [54]. The present study also includes 56.4% biopsy specimens to account for the problem of advanced tumour stages.

The fact that the patients examined in this study are mostly older men with a smoking history is consistent with the evaluation of large databases [4]. Surgery is the recommended treatment for early stage tumours, corresponding to our finding that surgery is an independent survival factor (*p* < 0.01, likelihood-ratio test, score-logrank test, Wald test). While surgical therapy is the treatment of choice in early tumour stages, options are limited for patients with advanced and metastatic tumours. The effect of chemotherapy in PSC is debated controversially. Although some studies demonstrate the efficacy of neoadjuvant and adjuvant chemotherapy [56,57], conflicting results have been found in advanced stage patients receiving systemic chemotherapy. Some authors report resistance to first-line chemotherapy [13,14,15], whereas in other studies chemotherapy shows a positive impact on overall survival, including an analysis of the SEER database of 1640 patients (hazard ratio, 0.78; 95% CI, 0.62 to 0.98) [57]. In our study, chemotherapy is an independent prognostic factor with a significant reduction in the risk of death for patients with PSC (*p* = 0.0021, score-logrank test) regarding the entire cohort, which also includes early tumour stages I–IIIA. These patients had surgery and received adjuvant radio- or chemotherapy. Considering only patients in advanced stages IIIB–IV, chemotherapy is not an independent prognostic factor in our study. This is in line with the studies referred to earlier, which suggest that PSC should be considered chemoresistant, at least in the context of palliative therapy. Great hopes are therefore placed in immunotherapy and targeted therapy. In recent years, both therapeutic approaches have revolutionized the treatment of NSCLC. As mentioned above, although there are already several studies on the immunological and molecular characteristics of sarcomatoid carcinomas, the data are sparse owing to the rare occurrence of PSC, and mostly only small cohorts or cohorts of different ethnicities were included.

Immunotherapy is based on the recognition that tumours are able to inhibit the T-cell-mediated immune response by expressing PD-L1 on the cell surface. Immune checkpoint inhibitors (ICIs) prevent PD-L1 from binding to the programmed cell death protein on T-cells, resulting in the destruction of tumour cells by a targeted cytotoxic response from CD8-positive T-cells. We find expression of PD-L1 in 147/179 (82.1%) of PSCs, and 106/179 (59.2%) of all tumours show high expression in at least 50% of tumour cells. This is quite comparable to the results of previously available studies with 13 to 75 patients included, demonstrating expression of PD-L1 in 53.3% to 89.4% of cases [18,19,21]. In other subgroups of NSCLC, PD-L1 expression in tumour cells is significantly lower, e.g., in adenocarcinoma (25.0%, 5/20), squamous carcinoma (15.8%, 3/19), or large cell carcinoma (20.0%, 3/15) [21]. The high proportion of PD-L1-expressing tumour cells in PSC suggests a promising therapeutic approach for the application of ICIs in patients with advanced tumour stages and possibly also for adjuvant or neoadjuvant therapy of sarcomatoid lung carcinomas in the early tumour stages. The anti-tumour efficacy of ICI has already been demonstrated in several studies with PSC patients in advanced tumour stages after first line chemotherapy [25,58]. PD-L1 expression is significantly positively associated with an improved response and prolonged PFS (*p* = 0.026 and *p* = 0.04, respectively) [18]. Of great interest in this context is that PSC with high PD-L1 expression (TPS ≥ 50) had one or more genomic alterations in 93/102 (91.2%) of cases. In 34/102 (33.3%), these were oncogenic driver mutations or gene rearrangements for which targeted therapeutics are already approved. The key question here is whether sarcomatoid carcinomas of the lung with high PD-L1 expression and treatable oncogenic driver mutations should receive immunotherapy or, according to current guidelines, targeted therapy first. Of great potential in this regard are also combined therapeutic approaches, for example by using small molecule immune checkpoint inhibitors. These agents target PD-1 and other negative checkpoint regulators such as the V-domain Ig Suppressor of T cell Activation (VISTA) and may offer advantages such as ease of dosing, the ability to control immune-related side effects due to their shorter pharmacokinetic exposure, and the capacity to affect more than one signalling pathway to improve efficacy [59,60]. In addition, cancer vaccines appear promising in preclinical studies for patients with PD-L1 expression and concomitant driver mutations. For example, Gus et al. recently showed that the newly discovered PD-L1 peptide vaccine for B cells (PDL1-Vaxx) exhibited a strong immune response and effective anti-tumour immunity in several syngeneic mouse models and worked synergistically in combination with a dual HER-2 B cell vaccine (B-Vaxx) [61,62].

Another mechanism by which tumour cells evade autoimmune cytotoxic defence can be explained by the processing of altered epitopes on the cell surface. Based on the fact that mutations give rise to neoantigens, these are presented on the tumour cell surface as short-range peptide fragments (epitopes) bound to HLA/MHC class I. The intracellular processing of epitopes is a complex process that includes proteasome degradation. It has been shown that mutations that change proteasomal cleavage patterns lead to a dysfunctional presentation of epitopes on the tumour cell surface. Consequently, the tumour cells are not recognised as “foreign” by the T-cell-mediated defence and are therefore not eliminated. As previously shown with a deep learning model, altered epitope processing occurs in NSCLC due to these processing mutations, and it has been demonstrated that this mechanism is independent of PD-L1 expression [39]. In addition, we could show that tumour cells with altered epitope processing are exposed to selection pressure under ICI therapy, which leads to the development of resistance and a deterioration in overall survival [49]. In the present study, we demonstrate that processing mutations are significantly more frequent in PSC than in other NSCLC types (*p* < 0.05, Fisher’s exact test) and that the majority of altered epitopes are no longer bound to the known MHC class I supertypes. The activation of T-cell function is therefore likely to be limited. In this collective, a high amount of processing mutations is linked to dismal survival (*p* < 0.05, likelihood-ratio test, score-logrank test, Wald test). Especially in advanced-stage cancer patients, the presence of processing mutations is mostly an independent prognostic factor for survival in patients with PSC (Figure 6B). This marks processing mutations certainly as influential factor, when focusing strictly on PSC. This can also be supported by the fact that patients who have received ICI show better overall survival when no or few processing mutations are present compared to patients who have many processing mutations (*p* < 0.05, likelihood-ratio test, score-logrank test, Wald test). Though, these results should be interpreted with caution, as of the ten patients treated with ICI, only four patients received monotherapies with pembrolizumab. The remaining patients received combined ICI and chemotherapy. Larger cohorts need to be studied to determine whether the presence of processing mutations has predictive value with regard to the possible development of resistance to ICI treatment. However, after pooling data from different NSCLC cohorts, it very much seems that the histological subtype is more influential than processing mutations, as SARC NSCLCs without processing mutations already display dismal survival (Figure 8). In contrast, processing mutations are a distinguish factor in the PSC subgroup with even worse outcome (Appendix A), indicating at their usefulness as a tool for patient stratification in advanced-stage PSC. This is specifically important as the presence or absence of processing mutations allows for therapy stratification independent of PD-L1 expression.

In addition to immunotherapy, the potential of targeted therapy is also of particular interest, as the response of PSC to conventional radio- and chemotherapy is uncertain, especially in the advanced tumour stages. Previous work has shown that a significant proportion of PSC has genomic alterations, with mutations most commonly detected in TP53, KRAS, MET, EGFR, BRAF, HER2 and RET [17,51,53,63,64]. In the largest cohort of the Western population studied to date, with 125 patients included, Schrock et al. found genomic alterations listed in the NCCN NSCLC guidelines in 30% of cases. The distribution and frequency of the mutations found are essentially consistent with the results of our study; we find driver alterations with currently approved treatment options in 35.4% of patients. However, our rate of EGFR mutated patients is lower (1.7% vs. 8.8%) and mutations in ERBB2/HER2 did not occur in our patients. In contrast to this study, we detect rearrangements in RET (1.7%; 3/175) and in ALK (2.3%; 4/175). Highly interesting and in agreement with the mentioned study above [51], patients with PSC show a higher proportion of MET exon 14 skipping events (12% resp. 13.7% in our study). This is significantly higher compared to other types of NSCLC patients, e.g., adenocarcinoma who only present with MET exon 14 skipping mutations in 2.3% [54]. MET is a receptor for hepatocyte growth factor, which is involved in the growth, invasion and metastasis of malignant tumours. It is therefore expected that patients with MET exon 14 skipping mutations should respond to targeted anti-MET therapy. There are already several tyrosine kinase inhibitors that have shown efficacy in clinical trials with NSCLC patients [65,66]. For PSC patients, however, there are only individual reports [67], and prospective studies are still pending. In a Chinese study with 70 NSCLC, 25 patients with PSC received the oral selective MET tyrosine kinase inhibitor savolitinib and showed an overall response rate of 40%, while 8 patients had a stable disease [68]. To confirm these achievements, a Chinese multicentre, open-label phase III trial is currently underway to assess the efficacy, safety and tolerability of savolitinib in NSCLC patients with MET exon 14 skipping mutations [NCT04923945].

Of note is a study by Mayenga et al. in which 6 NSCLC patients with MET exon 14 skipping demonstrated a remarkably long response to therapy with ICI. Among these patients was a patient with PSC, PD-L1 expression on tumour cells 40%, who received nivolumab after failure of first line chemotherapy. The tumour showed a rapid complete response that lasted 25 months. After discontinuation, the patient showed no signs of recurrence even more than 16 months later [69]. These results suggest an efficacy of therapy with ICIs in PSC patients with MET exon 14 skipping events. As mentioned above, patients with oncogenic drivers had no or few processing mutations and showed independent from PD-L1 expression a survival benefit from ICI therapy. The question is therefore whether immunotherapy or targeted therapy with selective MET kinase inhibitors is beneficial in PSC patients with no or few processing mutations and co-occurring oncogenic driver mutations, and whether an assessment of processing mutations can facilitate therapy decision-making.

In conclusion, pulmonary sarcomatoid carcinomas, which show more aggressive biological behaviour and poorer clinical outcome than other types of NSCLC, reveal a complex interaction of immunological characteristics and genomic alterations that offer potential opportunities for a precision oncology treatment approach. High Mutational Load and high expression of PD-L1 as well as other immune escape mechanisms such as a low amount of processing mutations, are prognostic and indicative for therapy with immune checkpoint inhibitors. Additionally, a variety of genomic alterations offer potential for the use of targeted therapy and require prospective clinical studies on response rates, efficacy and tolerability. In this context, it is essential to address the question of prioritizing immunotherapy or targeted therapy in the presence of PD-L1 overexpression and oncogenic driver mutations. Preclinical data on tumour vaccines showing synergistic effects in immune checkpoint regulation and targeting of driver mutations or the use of small molecule immune checkpoint inhibitors, which may allow capacities for combination with targeted therapies, are promising future treatment options. Whether processing mutations occurring frequently in PSC can be a suitable biomarker for selecting patients for ICI therapy independently from PD-L1 expression must be shown in further studies with larger patient cohorts.

## 4. Materials and Methods

### 4.1. Patients

From January 2006 to September 2022, 179 patients were primarily diagnosed with sarcomatoid lung cancer in the Department of Pathology, Lung Cancer Center, Helios Klinik Emil von Behring (HKEvB), Berlin, Germany. The Lung Cancer Center of the HKEvB has been certified annually by the German Cancer Society since 2009. The histological diagnosis was carried out by experienced pathologists according to the WHO criteria using the four-eye principle. For NSCLC patients who underwent primary surgery, the pathological tumour stages assigned according to the TNM staging system for lung cancer of the American Joint Committee on Cancer and the International Union for Cancer Control (UICC; 6th, 7th resp. 8th edition) were used in this study. All NSCLC patients who did not undergo primary surgery were classified according to the UICC clinical stages at the time of initial diagnosis. Tumour tissue was available for molecular pathology analysis from 175/179 (97.8%) patients. In the remaining four patients, molecular testing could not be performed because a sufficient amount of tumour tissue could not be obtained, or the tissue used provided too little DNA, or the DNA was of poor quality. For the present study, daily routine data from the specified period were retrospectively analysed. Genomic analysis of tumour tissue was approved by all participating institutions. Informed consent was obtained from all patients. The research protocols of two studies, which also included the genomic analysis of tumour tissue from some of these patients, were reviewed and approved by the Ethics Committee (Eth-X-AD/19 and Eth-48/20) of the Berlin Medical Association.

### 4.2. Nucleic Acid Extraction and Quantification

The tumour tissue fixed with 4% buffered formalin was macroscopically examined, macrodissected and embedded in paraffin. Two 20 µm thick unstained paraffin sections were made from each tumour. A 3 µm section was then made and stained with H&E to determine the percentage of tumour cell content. Tumour tissue was then marked using light microscopy and scraped from the unstained paraffin section.

Nucleic acid extraction was performed automatically according to the manufacturer’s instructions using the Promega Maxwell 16 FFPE PLUS LEV DNA Kit (Cat. #AS1135) or RNA Kit (Cat. #AS1260) on the Maxwell instrument (Promega, Madison, WI, USA). The concentration of extracted nucleic acids was determined using the QuBit^®^ dsDNA HS Assay Kit (Cat. #Q32851) or the Qubit^®^ RNA HS Assay Kit (Cat. #Q23852) in the Invitrogen Qubit 3.0. fluorometer (Invitrogen/Thermo Fisher Scientific, Waltham, MA, USA) as described in the manufacturer’s instructions (Cat #Q32851). DNA quantification was performed using the TaqMan^®^ RNAse P assay (Thermo Fisher Scientific, Waltham, MA, USA) according to the manufacturer’s instructions (Cat. #4316831). The cDNA synthesis from the extracted RNA was performed using a Superscript VILO cDNA Synthesis Kit (Invitrogen, Thermo Fisher Scientific, Waltham, MA, USA) in the SimpliAmp Thermalcycler (Thermo Fisher Scientific, Waltham, MA, USA) also according to the manufacturer’s instructions (Cat. #11754250).

### 4.3. Next-Generation Sequencing

Mutation status was determined by performing amplicon-based next-generation sequencing on the Ion Torrent S5 XL and Ion Torrent S5 Prime sequencing platforms (both Ion Torrent by Thermo Fisher Scientific, Waltham, MA, USA). NGS Libraries were generated according to the Ion AmpliSeq™ Library Kit 2.0 User Guide (Version E.0 MAN0006735). The Ion AmpliSeq™ Library Kit 2.0 (Cat. #4480441) was used with Ion XPress Barcode Adapters (Cat. #4474517) and SampleID (Cat. #4479790). Libraries were quantified using the Ion Library TaqMan™ Quanfication Kit (Cat. #4468802) according to the user’s manual instructions (MAN0015802), equalized and automatically loaded onto Ion 520™ (Cat. #A27762) or Ion 530™ Chips (Cat. #A27764) with a pool concentration of 35 pM using an Ion Chef instrument (Thermo Fisher Scientific, Waltham, MA, USA, Cat. #4484177).

The single-pool DNA Community Panel CLv2 (Thermo Fisher Scientific, Waltham, MA, USA, [70] includes the following genes: AKT1, ALK, BRAF, CTNNB1, DDR2, EGFR, ERBB2, ERBB4, FBXW7, FGFR1, FGFR2, FGFR3, KRAS, MAP2K1, MET, NOTCH1, NRAS, PIK3CA, PTEN, SMAD4, STK11, TP53. The CLv2 panel was used to determine the mutational status of 137 patients in total. The two-pool nNGMv2 lung panel was used to determine mutational status for the remaining 37 patients as of March 2020. It was developed and validated by the nNGM and revalidated regularly through proficiency-testing. It includes relevant parts of the following genes: ALK, BRAF, CTNNB1, EGFR, ERBB2, FGFR1, FGFR2, FGFR3, FGFR4, HRAS, IDH1, IDH2, KEAP1, KRAS, MAP2K1, MET, NRAS, NTRK1, NTRK2, NTRK3, PIK3CA, PTEN, RET, ROS1, STK11, and TP53. Samples were analysed by performing amplicon-based next-generation sequencing on the Ion Torrent S5 XL and Ion Torrent S5 Prime sequencing platforms (both Ion Torrent by Thermo Fisher Scientific, Waltham, MA, USA). In view of being able to compare the results of the two panels used, only genes that are present in both panels were evaluated: ALK, BRAF, CTNNB1, EGFR, ERBB2, FGFR1, FGFR2, FGFR3, KRAS, MAP2K, MET, NRAS, PIK3CA, PTEN, STK11, and TP53.

In order to investigate the influence on antigen processing, only non-synonymous mutations were included. The selection criteria were a coverage above 500 reads, a variant coverage above 20 reads and an allelic frequency above 3 percent, but below 90 percent, thereby excluding fixation artifacts. Assuming mutations leading to altered antigen processing were a prominent occurrence, only variants with a tumour allelic frequency above 25% were selected for further investigation.

Fusion analysis was performed on the Ion Torrent S5 XL and Ion Torrent S5 Prime sequencing platforms (both Ion Torrent by Thermo Fisher Scientific, Waltham, MA, USA) using the amplicon-based Oncomine Focus RNA assay (Cat. #A35956). The assay comprises 284 different fusion transcripts, including the genes ALK, AXL, BRAF, EGFR, FGFR1, FGFR2, FGFR3, MET, NTRK1, NTRK2, NTRK3, PAX8, RAF1, RET, ROS1, TMPRSS2, and others (for the complete list of all 284 fusions, see Appendix A).

### 4.4. Variant Calling and Classification

Bioinformatics pipelines in Torrent Suite 5.12, OS Ubuntu 14.04, were used for DNA coverage analysis and variant calling. Variant classification was performed with annovar. RNA sequences were uploaded from Torrent Suite to IonReporter Server 5.10, and RNA fusion analysis was performed using the former. All data were transferred electronically to a laboratory information management system (ionLIMS, Heidelberg, Germany).

### 4.5. Processing Escape Analysis

#### 4.5.1. Demographic Data and Study Design

For comparison purposes two NSCLC cohorts, whose patients displayed no tumours with sarcomatoid differentiation, were utilized. The first cohort established included 48 patients [49,50]. Subtypes were roughly distributed equally: lung squamous carcinomas (LUSCs), *n* = 25 and lung adenocarcinomas (LUADS), *n* = 23. All patients were diagnosed in advanced tumour stages IIIB–IV. First-line treatment consisted of either chemo- or radiation therapy. Chemotherapeutics were applied as a combination of two agents. One component always contained either cis- or carboplatin. The platinum-component was combined with gemcitabine, vinorelbine or pemetrexed. Twenty-eight patients additionally received radiation treatment. Immunotherapeutics, i.e., nivolumab, were not administered in first-line treatment. However, all patients received it at some point in their treatment history. PD-L1 expression was quantified in most patients, primarily by utilizing the tumour proportion score (TPS), i.e., PD-L1-positive membrane staining on tumour cells. Throughout this work, this cohort will be termed as ICB cohort.

The second cohort consisted of data obtained from The Cancer Genome Atlas (TCGA). Clinical and sequencing data (whole-exome sequencing) of 408 lung cancer patients (230 LUADs, 178 LUSCs) were available [71,72]. The validation cohort also includes patients with stage I-II disease who were preferentially treated with surgical resection and adjuvant radio- or chemotherapy. Throughout the rest of this work, this cohort will be termed as TCGA cohort.

#### 4.5.2. Annotation of Genomic Variants on the Protein Level

The use of our established [49,50] bioinformatic workflow necessitates information on how mutations are characterized at the protein level. We mainly focused on non-synonymous mutations causing missense mutations. VCF files obtained after sequencing procedures were filtered as mentioned in 4.3. The filtered results were used as the input for The Ensembl Variant Effect Predictor (VEP) [73]. The VEP output was then again filtered for non-synonymous mutations, i.e., missense variants. It should also be noted that we also considered different transcript variants for downstream analysis.

#### 4.5.3. Exploratory Data Analysis

All forms of downstream exploratory analysis were performed in the R programming environment (The R Foundation for Statistical Computing, Institute for Statistics and Mathematics, Vienna, Austria; v. 4.2.2).

##### Correlating Non-Synonymous Mutations to Epitopes

In the first step, we predicted all potential epitopes, divided by affinity to specific MHC class I molecules, that may be generated around the area of the protein, where one specific missense mutation is located. The predictive models, utilized for this purpose, work of an amino acid similarity matrix applied to MHC interactions. This method was originally devised by B. Peters and colleagues [74]. The epitope sequence and its two flanking regions towards the C- and N-terminus were used as an input for the next procedure. Flanking regions were included since they yield potential influence on epitope processing due to their sidechain interactions [75,76].

##### Predicting Proteasomal Epitope Processing

Proteasomal cleavage of epitope sequences was predicted by NetChop 3.1. It is based on the principle of convolutional neural networks. For this purpose, NetChop provides different algorithms, which differ by the training set for the prediction model. These algorithms are NetChop Cterm and NetChop 20S. One was trained on established in vivo MHC class I ligand structures (Cterm), while the second was trained based on in vitro-generated proteasomal cleavage data (20S) [77,78]. Cterm was selected as the primary method for this study.

For amino acids within the epitope and its flanking regions, a cleavage probability is calculated. NetChop was executed two consecutive times with wildtype and mutated epitope fragments. Altered epitope processing was defined as a 50% difference in cleavage probability between the wildtype and mutated epitope.

##### Verification of MHC Class I Binding and Immune Activation

NetMHC (Version 4.0) was used to predict MHC binding properties of altered epitopes [79,80]. Like NetChop, NetHMC is also based on convolutional neural networks. The algorithm contains data about binding affinities of epitopes towards different MHC class I types. High affinity is reflected by low IC50 values. For this study, we focused on the binding affinity between altered epitopes and 12 prominent MHC class I supertypes [81,82]. Like NetChop (2.5.2), NetMHC was run two consecutive times to compare unmutated and mutated epitopes.

The immune activation potential of wildtype and mutated epitope sequences was evaluated by the Class I Immunogenicity tool. This tool is offered as a resource for research purposes by the Immune Epitope Database (IEDB) [83]. For each epitope, a score is returned that reflects the potential of TCR activation. Should one epitope exceed the average immunogenicity score minus 2× standard deviation of its wildtype correspondent, it was defined as a TCR-activating epitope.

##### Statistical Analysis

For any metric variable of interest, data were fitted to analysing general normal distribution by Shapiro–Wilk-test [84]. Depending on the result, a non-parametric test (Wilcoxon Mann–Whitney rank sum test) or parametric (students *t*-test) was applied when correlating the variable of interest with to an ordinal variable, which included exactly two groups [85]. If one ordinal variable contained more than two groups, the non-parametric Kruskal–Wallis test or parametric Analysis of Variance (ANOVA) were utilized. Significant differences between double dichotomous categorical variables were evaluated by Fisher’s exact test. In case one variable included more than two groups, the Pearson’s Chi-square test was applied.

Overall survival (OS) was calculated for each patient. Patient survivability was calculated by Cox proportional hazard (CoxPH) models. Model accuracy could be inferred from the score log-rank test, Wald-test and likelihood-ratio test. Group-based survival differences were visualized by Kaplan–Meier plotting. *p*-values may vary after repeating statistical tests multiple times, which necessitates *p*-value adjustment by false discovery rate (FDR). The level of statistical significance was set at *p* ≤ 0.05 after FDR-adjustment.

### 4.6. Immunohistochemistry

Immunohistochemical analyses for PD-L1 expression were performed on all 179 PSC samples. For PD-L1 immunoreactions, 3 µm thick sections of FFPE tumour samples were prepared and mounted on Superfrost™ Plus adhesion microscope slides (Epredia, Breda, the Netherlands). PD-L1 expression was visualised using the automated BOND system in combination with BOND Polymer Refine Red Detection (Leica Biosystems Newcastle Ltd., Balliol Business Park, Benton Lane, Newcastle Upon Tyne, UK) according to the manufacturer’s instructions. Briefly, samples were dewaxed using BOND Dewax Solution (Leica, cat#AR9222), run through a descending alcohol series and washed using BOND Wash Solution 10X Concentrate (Leica, cat#AR9590). The BOND Wash Solution 10X Concentrate is diluted 1:10 in the instrument before use. Pre-treatment is carried out with BOND Epitope Retrieval Solution 2 (“BOND ER Solution 2”) (Leica, cat#AR9640) for 30 min at 100 °C. After a wash step, the antibody Programmed Death Ligand 1 (Clone 73-10) (Leica, cat#PA0832) is bound for 20 min at room temperature. Programmed Death Ligand 1 (73-10) primary antibody is optimally diluted for use on the BOND system, and reconstitution, mixing, dilution or titration of this reagent is not required. The staining was performed with the BOND Polymer Refine Red Detection (Leica, cat#DS9390). Sections were then manually dehydrated using an ascending alcohol series (70%, 96%, abs, abs, xylene, xylene, xylene) and coverslipped. PD-L1 protein expression on tumour cells was determined using the Tumour Proportion Score (TPS). No staining or up to 1% partial or complete staining of tumour cell membranes of any intensity was scored as 0, 1–49% stained tumour cells were scored as 1+ and 50% or more stained tumour cells were scored as 3+ (high) PD-L1 expression. For quality assurance of the antibody, we participate annually in the QUIP proficiency testing (Quality Assurance Initiative Pathology GmbH, audited by the European Society of Pathology).

## Figures and Tables

**Figure 1 ijms-24-10558-f001:**
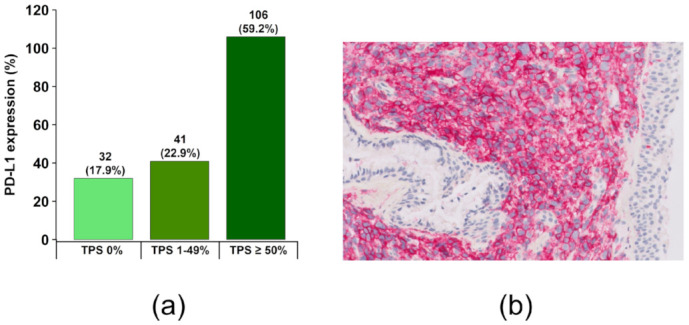
PD-L1 expression in sarcomatoid NSCLCs. (**a**) PD-L1 expression in relation to the percentage of membrane-stained tumour cells, indicated as tumour proportion score (TPS score). (**b**) Membranous expression of PD-L1 in sarcomatoid tumour cells, note the lack of expression in surface and bronchial epithelium (200× magnification). PD-L1 immunostaining was performed using a (73-10) monoclonal antibody.

**Figure 2 ijms-24-10558-f002:**
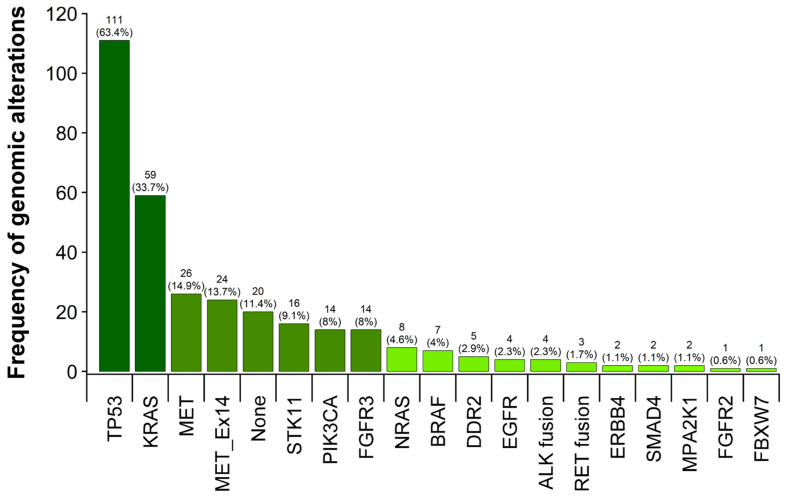
Frequency of genomic alterations in 175 primary diagnosed sarcomatoid lung carcinomas. A high frequency is presented by darker colour grading. Alterations in the MET oncogene leading to alternate transcripts missing regulatory domains in exon 14 are listed separately (MET_Ex14).

**Figure 3 ijms-24-10558-f003:**
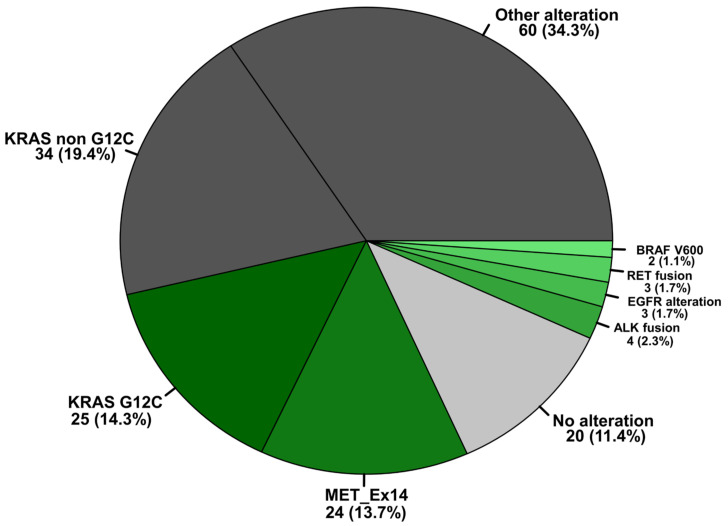
Distribution of therapy-relevant genomic alterations in PSC. The proportion of mutations and gene re-arrangements for which there are currently approved targeted therapies are highlighted in shades of green, depending on their abundance within the sarcomatoid (SARC) NSCLC cohort (61/175; 34.9%). The proportion of additional genomic alterations without a current targeted treatment option are outlined in dark grey (94/175; 53.7%), and tumours without genomic alterations are shown in light grey (20/175; 11.4%).

**Figure 4 ijms-24-10558-f004:**
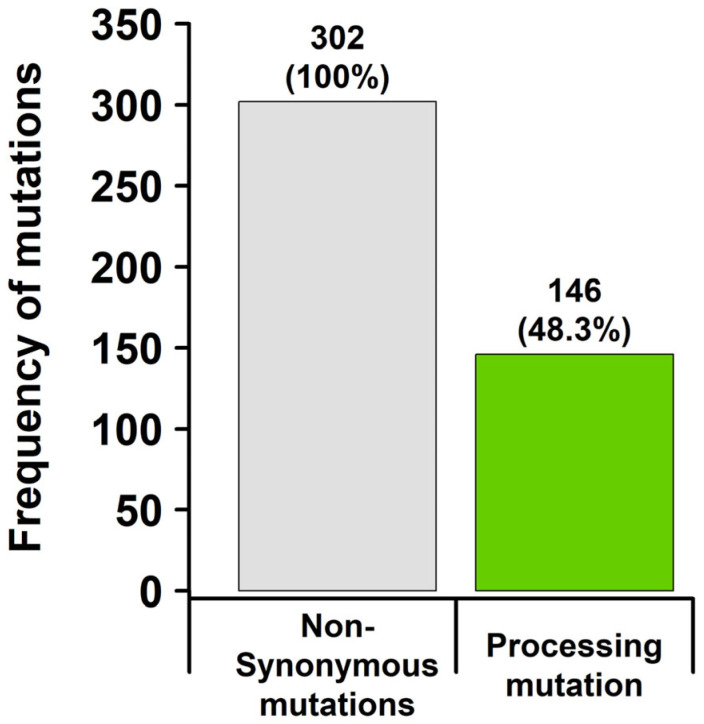
Total amount of non-synonymous mutations (missense mutations) and processing mutations identified in the SARC NSCLC cohort. These mutations were identified in 145 patients. Overall, 146/302 (48.3%) of those mutations lead to differently processed epitopes with less MHC representation.

**Figure 5 ijms-24-10558-f005:**
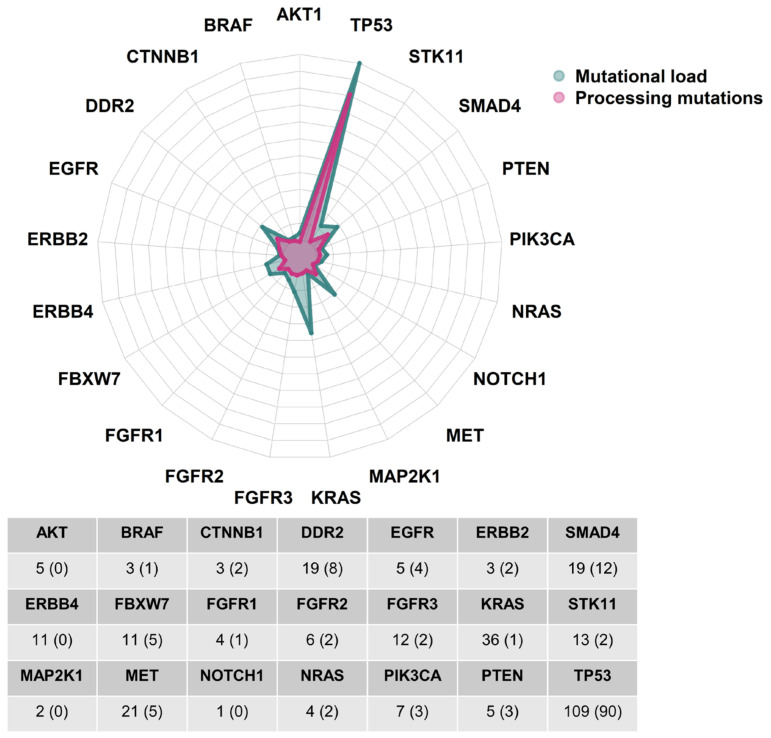
Distribution of total missense mutations and processing mutations across genes in SARC NSCLCs. The total distribution of missense mutations is displayed in green, while the total distribution of processing mutations is displayed in red. A frequency table, showing mutation count per gene, is also provided below the plot.

**Figure 6 ijms-24-10558-f006:**
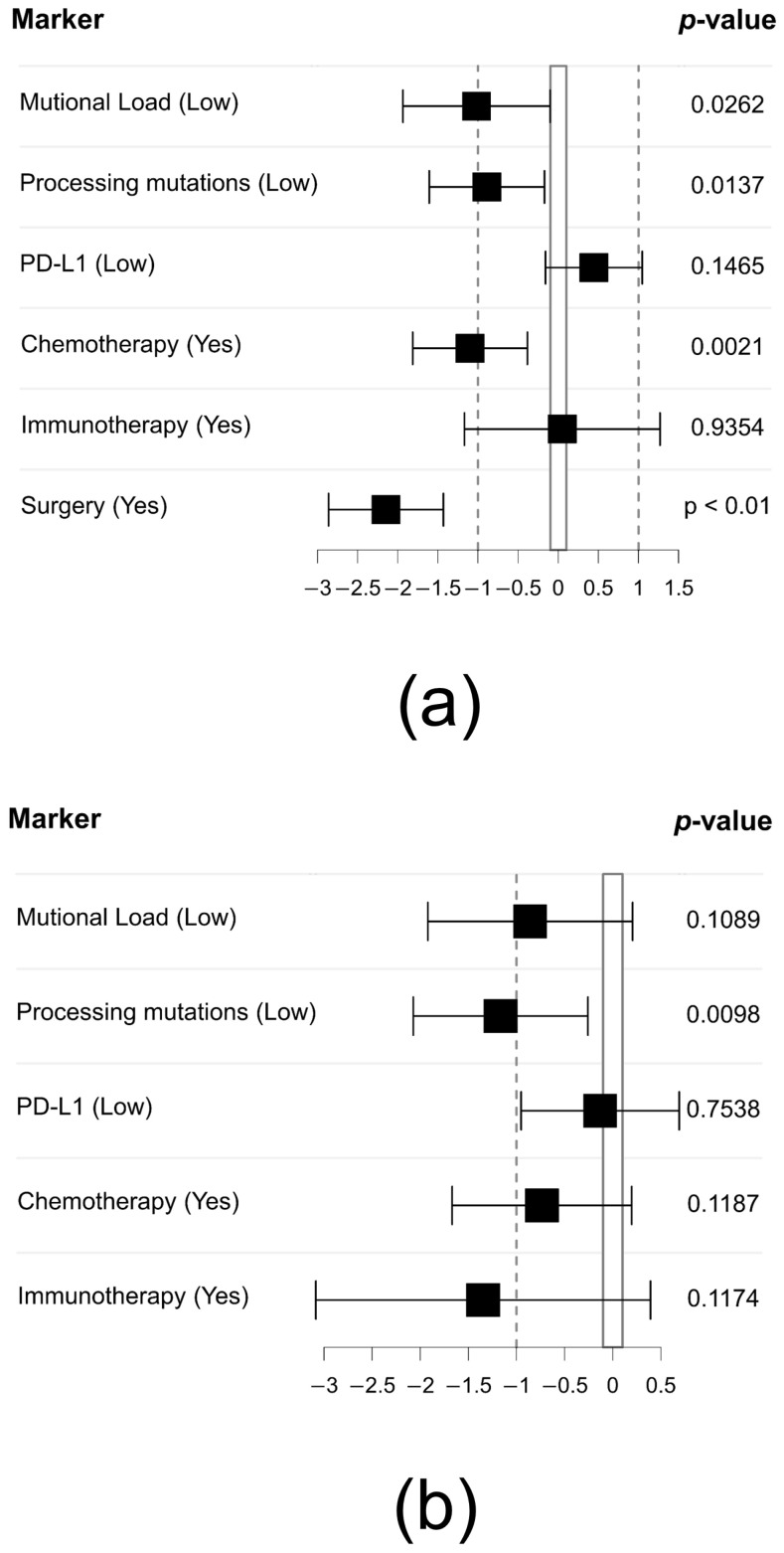
Correlation of various covariates with OS and display of hazard ratios. (**a**) Cox proportional hazard models were utilized to estimate the risk of death (hazard ratio) throughout the entire cohort based on Mutational Load (total amount of missense mutations), processing mutations (high/low), PD-L1 expression as well as various forms of therapy. Surgery, the application of chemotherapy. A low amount of non-synonymous mutations and processing mutations as well as the application of chemotherapy or surgery were linked with a significantly reduced risk of death. (**b**) Cox proportional hazard models were applied only on advanced-stage patients (stage IIIB–IV). Surgery was omitted from the analysis, due to advanced-stage patients not receiving surgery as a primary form of therapy. Processing mutations retain their significant influence on OS. The *p*-value was calculated by score-logrank test.

**Figure 7 ijms-24-10558-f007:**
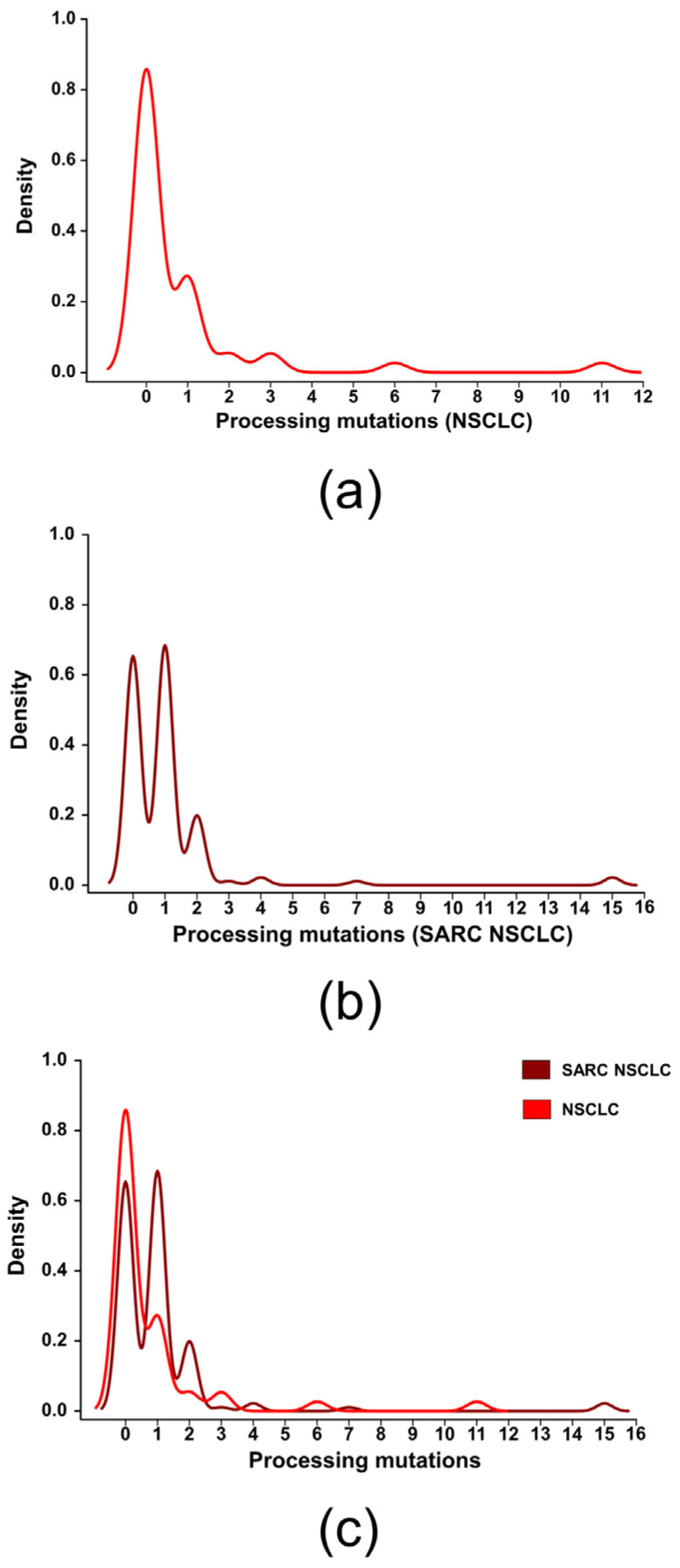
Frequency of non-synonymous mutations leading to altered epitope processing in various NSCLC subsets. (**a**) Density distribution of samples (ICB cohort) across specific mutation counts (processing mutations) in the ICB cohort (PMID: 32922086, 34996407). (**b**) Density distribution of samples (SARC NSCLC cohort) across specific mutation counts (processing mutations). (**c**) Density distribution of samples (both NSCLC cohorts) across specific mutation counts (processing mutations).

**Figure 8 ijms-24-10558-f008:**
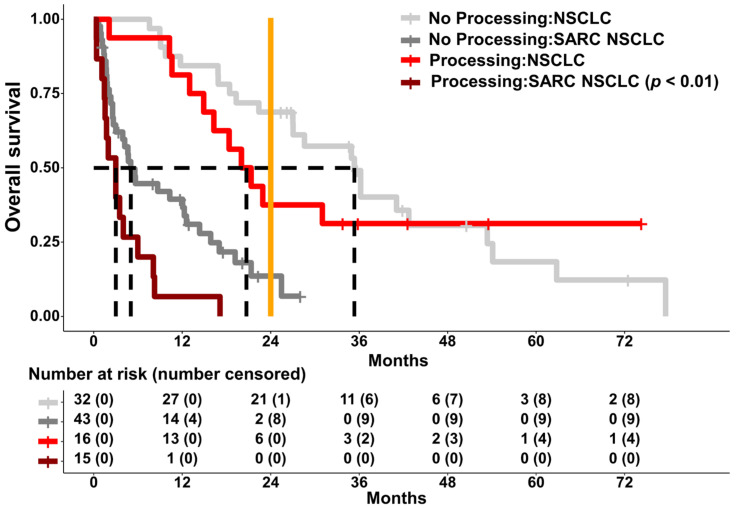
Kaplan–Meier plots displaying correlations of processing mutations to OS in different NSCLC cohorts. A high mutation count constitutes more than one mutation. Dotted lines represent the median OS of each group. To focus on the course of survival, influenced by processing mutations, in different NSCLC cohorts, the other graphs/groups are greyed out. The *p*-value was calculated by score-logrank test.

**Table 1 ijms-24-10558-t001:** Baseline patient characteristics.

	No.	%
Age	179	100
≤60	54	30.2
>60	125	69.8
Gender	179	100
Men	105	58.7
Women	74	41.3
Histological subtype	179	100
Pleomorphic Carcinoma	116	64.8
Spindle Cell Carcinoma	42	23.5
Giant Cell Carcinoma	16	8.9
Carcinosarcoma	5	2.8
Pulmonary Blastoma	0	0.0
Histological Grading	179	100
G1	0	0.0
G2	1	0.6
G3	173	96.6
G4	5	2.8
Stage TNM as available	177	98.9
IA1	4	2.3
IA2	3	1.7
IA3	3	1.7
IB	13	7.3
IIA	7	4.0
IIB	24	13.6
IIIA	33	18.6
IIIB	16	9.0
IIIC	1	0.6
IVA	50	28.2
IVB	23	13.0
Smoker status as available	135	75.4
Never smoker	20	14.8
Former smoker	59	43.7
Current smoker	56	41.5

**Table 2 ijms-24-10558-t002:** Genomic alterations with approved therapy options in PSC presenting with high PD-L1 expression (TPS ≥ 50%).

	No.	%
PSC with PD-L1 ≥ 50%	106	59.2
DNA/RNA available	102	96.2
≥1 mutation or fusion	93	91.2
None	9	8.8
KRAS G12C	19	18.6
MET Ex14skip	12	11.8
BRAF V600E	1	1.0
CCDC6-RET fusion	1	1.0
EML4-Alk (E13A20) fusion	1	1.0

## Data Availability

The datasets analysed in this study are available from the corresponding author upon request.

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
