# Peer review of "Integrated Clinical, Molecular and Immunological Characterization of Pulmonary Sarcomatoid Carcinomas Reveals an Immune Escape Mechanism That May Influence Therapeutic Strategies"

_ijms, 2023, doi:10.3390/ijms241310558_

Round 1

Reviewer 1 Report

  • This review outlines immunological and molecular features of PSC and ways to explore different immune escape mechanisms. Overall review is nicely designed and presented in an organized manner.
  • Comparision with respect to PSC is performed well. PSC and their histological subtypes are explained in good detail.
  • Various recent advances in immunotherapies are clearly explained, detail description is provided in general and gradually going into specific details.
  • Each section is demonstrated with adequate details.
  • Appropriate references were cited.
  • Abbreviations elaboration or glossary in beginning is recommended to remind the reader in the beginning of the article. 
  • Indent change is recommended for figures to maintain consistency throughout the review with the text.
  • Conclusion part can be little more specific with additional detailing with respect to the future directives. 
  • doi's are missing for the references 1,5,8,10,63,66

Reviewer 2 Report

Dear,

The current study nicely analyzes the prognostic factors affecting the overall survival in a big cohort of pulmonary sarcomatoid carcinoma (PSC) patients. You also compare their observations to the previously published NSCLC cases which helps the reader to comprehend the seriousness and the refractory nature of PSCs.

The paper is well-written and discussed. I have only a few minor comments regarding to the study design and the paper:

General comments:

1-      Majority of the patients are Grade-3. Does this really reflect the PSC patients where the diagnosis/treatment option decision is made? If not, how do you generalize your results? Because undifferentiated cells prone to divide more often, and have higher mutational ratios, does it make your conclusions biased towards the high-mutational burden subgroup of PSC cases?

2-      Just for the reader please define clearly what you mean by “processing mutations”?: Mutations affect the proteasomal machinery subunits that change the end-product peptides, or the subunits/parts of the proteins that are cleaved by the proteasomal system?

In your case I believe it is the later: are there any reports showing the importance of these subunits in the immune control of PSC cases: specific T-cell receptor, exhausted population, CART treatment candidates, etc.

3-      Regarding the immune evasion effect of the processing mutations: Most mutations are detected in the genes that can also drive the tumorigenesis. Is it a true-immune evasion effect caused by the loss of CD8 T cell recognition, or is it the mutated gene product that drives the tumor cell formation? Without supporting reports or functional data, one may find the immune evasion hypothesis and conclusion a bit too ambitious. At least it is worth discussing the additional tumorigenesis effects of the mutated genes.

Minor comments:

Line 67: even though it is general knowledge, -N2 involvement-, explain the term before using the acronym.

Line 84-94: Transcriptional, post-transcriptional, posttranslational mechanisms resulting in deficient MHC I presentation are more commonplace than genetic/genomic alterations. Maybe good to mention them briefly as well. 

Reviewer 3 Report

1.       The results are of great interest to the scientific community, primarily because the study examined a rare form of NSCLC.

2.       The figure 1 can be improved according to IJMS standards

3.       The figure 2 is in low resolution and should be improved

4.       In table 2, the proportion of positive tumors for PD-L1 is missing.

5.       Figures 4 and 6 could be improved – low resolution and quality

6.       Figure 8 – Sarcomatoid cohort would be better named as SARC NSCLC and indicated in the figure caption.

7.       It would be interesting to compare the mutational landscape (processing and non-processing variants) between PSC smokers and non-smokers.

Minor editing of English language required

Reviewer 4 Report

Effectively selecting patients that might well respond to immunotherapy is very important to make proper treatment strategies to enhance therapeutic efficiency and prolong patients’ survival rates. PD-L1 is one of the key factors associated with tumor metastasis. It is an important topic and interesting. However, the manuscript does not very well prepared. Some figures can be combined as one, but the images and the protocols are not clear. The followings are some concerns and comments have been pointed out that the authors may want to consider.

1.      Line 24: Please use italic p as it refers to a p-value throughout the manuscript.

2.      Line 31: I’d suggest the authors rephrase “one third (35.4%)”. For example, line 126, “more than one third” etc.

3.      Line 32: I’d suggest the authors use the “actual number + percentage” format. For example, 10% (5/50).

4.      Line 128 Table 1: a) Age (years old); b) Upper case “Men”, “Women”; c) Why in the “smoker” section only 135 patients? I’d suggest the authors include the left 44 patients as “unclear history” etc. d) Please include statistical method information and necessary abbreviations in the table note.

5.      Line 138 Figure 1 related: Please provide PD-L1 expression data. Immunostaining images or whatever else.

6.      Line 151 Figure 2 related: Please a) provide solid result/data instead of only frequency (%). b) provide a high-resolution image.

7.      Line 165 Figure 3 related: Please see comments to Figure 2.

8.      Line 193 Figure 4 related: Please see comments to Figure 2.

9.      Line 206 Figure 5 related: Please see comments to Figure 2.

10.  Line 216: Please be consistent with or without a space before and after the sign. For example, “=”.

11.  Line 228 Figure 6 related: Please see comments to Figure 2.

12.  Line 252 Figure 7 related: Please provide higher-resolution images and make explanations to make it clear. For example, what is the X-axis?

13.  Line 276 Figure 8 related: Is it possible to use the month as the X-axis?

14.  Line 451 materials and methods section: a) Please provide the cat# for all the reagents used in this study. b) Please specify the statistical method information for each analysis. For example, one-way ANOVA, and so on. c) Provide more protocol details for each method. For example, antibody dilution ratio, and so on.

15.  I’d suggest the authors discuss the potential combination and checkpoint therapy of small molecules. PMID: 35646678, PMID: 36211807, PMID: 34103659, etc. These are only recommendations and suggestions. Please check other literature by yourself.

16.  Please be consistent with the format throughout the manuscript.

Language editing is recommended

Reviewer 5 Report

The manuscript presents the results on pulmonary sarcomatoid carcinoma. It is an interesting study, however, there are several major concerns:

1. The section "Results" is not appropriate for publication. All figures are presented inappropriately. All Figures are missing error bars, it is also not clear what controls are for each of the Figure. Some Figures are hard to read.. Authors should present the Figures using professional scientific style..

2. The title of the manuscript is confusing. Title tells a reader that there is  immunological insight, however, authors only assessed PD-L1 expression and  did not show any immune escape mechanism..

Authors should carefully proofread the manuscript  as there are some mistakes throughout the manusctipt,  for example in Abstract "Pulmonary sarcomatoid carcinoma have ..." 

Round 2

Reviewer 4 Report

Thank you for the updates. I do not have further comments now. Good luck.

It's OK. Minor editing is recommended.

Reviewer 5 Report

The authors have addressed all the comments